# Surgery as an Emotional Strain: An Observational Study in Patients Undergoing Elective Colorectal Surgery

**DOI:** 10.3390/jcm11102712

**Published:** 2022-05-11

**Authors:** Ann-Kathrin Lederer, Ines Manteufel, Agnes Knott, Lampros Kousoulas, Paul Georg Werthmann, Maximilian Andreas Storz, Roman Huber, Alexander Müller

**Affiliations:** 1Center for Complementary Medicine, Department of Medicine II, Medical Center—University of Freiburg, Faculty of Medicine, University of Freiburg, 79106 Freiburg, Germany; i.manteufel@googlemail.com (I.M.); aggip@freenet.de (A.K.); paul.werthmann@uniklinik-freiburg.de (P.G.W.); maximilian.storz@uniklinik-freiburg.de (M.A.S.); roman.huber@uniklinik-freiburg.de (R.H.); alexander.mueller@uniklinik-freiburg.de (A.M.); 2Department of General, Visceral and Transplant Surgery, University Medical Center of the Johannes Gutenberg University, 55131 Mainz, Germany; 3Center of Surgery, Department of General and Visceral Surgery, Medical Center—University of Freiburg, Faculty of Medicine, University of Freiburg, 79106 Freiburg, Germany; lampros.kousoulas@uniklinik-freiburg.de; 4Institute for Applied Epistemology and Medical Methodology, University of Witten/Herdecke, 79110 Freiburg, Germany; 5Research Group Integrative Medicine, Department of General and Visceral Surgery, University Hospital Ulm, 89081 Ulm, Germany

**Keywords:** emotions, surveys and questionnaires, distress, treatment outcome, colorectal surgery, postoperative complication

## Abstract

Recent research suggests an impact of psychological distress on postoperative outcomes in orthopedic and neurosurgery. It is widely unknown whether patients’ mood might affect the postoperative outcome and complication rate in colorectal surgery. Over a period of 22 months, a monocentric, observational study among patients undergoing elective colorectal surgery without the creation of an ostomy was conducted. Patients were asked to fill in a standardized multi-dimensional mood questionnaire (MDMQ) preoperatively as well as on the third, sixth, and ninth postoperative days to assess mood, wakefulness, and arousal. The results of 80 patients (51% male, mean age 59 years) were analyzed. Almost half of the patients (58%) developed postoperative complications according to the Clavien–Dindo classification (Grade I 14%, Grade II 30%, Grade III 9%, Grade IV 3%). Patients’ mood increased continually from the preoperative day to the ninth postoperative day. Patients’ wakefulness decreased initially (pre- to third postoperative day) and increased again in the further course. Patients’ arousal decreased pre- to postoperatively. Neither preoperative mood, nor arousal or wakefulness of patients showed a clear association with the development of postoperative complications. In conclusion, preoperative psychological distress measured by MDMQ did not affect the postoperative complication rate of patients undergoing elective colorectal surgery.

## 1. Introduction

For decades, it has been known that surgery is a massive stress event for the human body, requiring an effective pain and stress reduction for rapid healing [1,2,3]. Surgery does not only imply physical strain, as it is also psychological distress due to a loss of control and even fear of death. The role of mental well-being in human health is emphasized by the World Health Organization’s definition of health as, “health is a state of complete physical, mental and social well-being” [4]. Stressful emotional events are able to provoke severe and life-threatening symptoms such as the broken heart syndrome [5]. Psychological distress can deteriorate the process of wound healing [6].

Anxiety, depression, and distress have been found preoperatively in up to 20% of surgical patients [7,8,9,10,11,12,13]. Preoperative psychological distress might be a potential risk of developing long-term psychopathology after surgery [7,8,9,10,11,12,13]. Recent research suggests that preoperative psychological distress might worsen patients’ recovery and outcome after orthopedic surgery [14]. Furthermore, preoperative depression has been associated with an inferior outcome for patients after different types of neurosurgery [15,16]. An association between the mental set of patients and the postoperative pain management in lumbar fusion surgery is hypothesized [17]. For major abdominal surgery, multimodal concepts for the physical and mental preparation of patients showed an improvement in emotional as well as functional outcomes after surgery [18,19]. Psychological prehabilitation before cancer surgery is suggested to have an impact on the recovery of patients [20]. Not only quality of life, but also survival and disease recurrence of patients with cancer appears to be affected by pre- and perioperative psychological distress in general [21,22,23,24]. Recovery protocols such as ERAS set the value on active patient education and engage a patient to be part of the therapy leading to an improved outcome [25,26]. It is assumable that the active role of patients decreases the loss of control, potentially reducing a feeling of being at the surgeon’s mercy and lowering psychological distress. In colorectal surgery, patients are faced with a special and shaming emotional strain as colorectal surgery is often related to the existence of an ostomy or to stool incontinence, respectively, which relevantly might affect mental well-being. To date, less is known about the pre- and postoperative psychological distress of patients undergoing colorectal surgery. Therefore, the study aimed to estimate the pre- and postoperative mood of patients undergoing elective colorectal surgery by a multi-dimensional mood questionnaire (MDMQ) and to evaluate the impact of mood on the development of postoperative complications in colorectal surgery hypothesizing that lower preoperative mood scores in MDMQ might be associated with the development of postoperative complications.

## 2. Materials and Methods

Between April 2018 and February 2020, a monocentric, paper-based, observational study among in-patients undergoing colorectal surgery at the Department for General and Visceral Surgery, University Medical Center of Freiburg was conducted. Patients were asked to fill in a multi-dimensional mood questionnaire preoperatively as well as on the 3rd, 6th, and 9th postoperative days to assess their psychological distress during their hospital stay.

The study was approved by the local ethical committee (EK-FR: 535/17) and was registered in the German Clinical Trial Register (DRKS00014059). The study was performed according to the principles of the declaration of Helsinki and the guidelines of ICH for good clinical practice (GCP). Written consent was obtained from all patients before onset.

### 2.1. Criteria of Inclusion and Exclusion

Eligible for inclusion were all adult patients undergoing elective colorectal surgery (left hemicolectomy, right hemicolectomy, sigmoid resection, deep anterior rectal resection, restoration of continuity) without primary intended creation of ostomy. All eligible patients had to be able to speak and understand German, sign informed consent, and fill in the questionnaire on their own. To avoid bias, patients undergoing emergency surgery were not considered. Patients who received ostomy were excluded as it was likely that the creation of ostomy would affect results of questionnaire, making it not comparable with patients without ostomy. Patients with acute psychiatric disorders (requirement for an acute therapy) were not eligible for participation.

### 2.2. Questionnaire

Patients’ moods were captured by the German version of the multi-dimensional mood questionnaire (MDMQ) [27]. The MDMQ is a commonly used and well-validated questionnaire in Germany to capture the mental state of a respondent [27,28,29,30,31]. It consists of 24 items, each with a five-point rating scale, to measure three bipolar dimensions of the current psychological state: good vs. bad, awake vs. tired, and calm vs. nervous. The results range between 1 (feeling bad, tired, nervous) and 40 points (feeling good, awake, calm) for each dimension, summing up to an overall maximum count of 120 points (maximum scores for feeling good, awake, and calm) for the whole questionnaire. Patients received the questionnaire by the study staff and were asked to fill it in independently and on their own and to return it to a nurse or the study staff after finishing. Questionnaires were only handed out during the hospital stay. The questionnaire is only available in German. The decision to measure mood by MDMQ was driven by the necessity of a standardized, but simple and quick-to-answer questionnaire, to ensure that postoperative patients are not additionally stressed by the questionnaire and are able to finish it in a few minutes. The results were evaluated by three authors (AM, AK, AKL) with the help of an evaluation template (Testzentrale, Göttingen, Germany).

### 2.3. Outcomes

Psychological distress was measured in three dimensions by the aforementioned questionnaire: mood (good vs. bad), wakefulness (awake vs. tired), and arousal (calm vs. nervous). Descriptive patient data and data of the postoperative course were captured with the help of the electronic patient file. Postoperative complications were assessed according to the Clavien–Dindo classification [32,33] with help of the extended Clavien–Dindo classification by Katayama et al. [34]. All complications were captured up to the 14th postoperative day.

### 2.4. Statistical Plan

The primary target was the relation between preoperative MDMQ result (“mood”) of patients and the rate of postoperative complications according to the Clavien–Dindo classification [32,33]. Secondary analyses comprised relation of postoperative MDMQ results and the rate of postoperative complications as well as the comparison of MDMQ results by time. Sample size was planned for a pilot trial as effect size was unknown. Considering a statistical power of 80% and a hypothesized medium effect size of 1 standard deviation, it was calculated that 36 participants would be needed to detect a statistical difference of *p* < 0.05 between the patients with and without postoperative complications (G*Power, Kiel, Germany). At least 22 (+30%) additional patients were included as a reserve for dropouts, leading to an overall sample size of 94 patients for evaluation of the primary target. Data were entered blinded for complication rates. Complications were added last to the data by a blinded member of the study team.

Statistical analysis was performed using IBM SPSS (version 27.0). The significance level was set to two-sided α = 0.05. Results were checked for normal distribution. Depending on distribution, evaluation was carried out by independent *t*-test/Mann–Whitney U Test for independent samples as well as by paired *t*-test/Wilcoxon Test for paired samples. Chi-squared test and Fisher’s exact test were utilized to test for trends and significance and compare groups of categorical data. Pearson’s correlation coefficient was used to measure the relationship between variables. Missing values were not completed. Patients with missing data were excluded from comparative analysis.

## 3. Results

The flow of participation is shown in Figure 1. Out of 96 eligible patients, 95 consented to participate. Out of these, 87 were able to complete the preoperative questionnaire precisely and return the questionnaire in time. Eighty patients completed the questionnaire preoperatively and at least day 3 postoperatively, leaving these patients for comparison of the preoperative result and the result on third postoperative day (POD).

Of the remaining 80 patients, 51% were male (*n* = 41) and 49% were female (*n* = 39). Patients were on average 59 years old (range 18–84 years). The most common type of surgery was right hemicolectomy (38%), followed by the removal of the sigmoid colon (35%). Further descriptive patient data is shown in Table 1.

### 3.1. Results of Multi-Dimensional Questionnaire (MDMQ)

The course of pre- to postoperative mood, wakefulness, and arousal is shown in Table 2 and Figure 2. The average score of mood (bad vs. good) increased continually from the preoperative day to POD9 (POD3 29.6 points to POD 6 31.1 points: *p* = 0.013, other POD comparisons did not reach the level of statistical significance). The average score of wakefulness (awake vs. tired) decreased significantly from the preoperative day to POD3 (24.1 to 28.8 points, *p* = 0.013) and increased significantly from POD 3 to POD 6 (25.3 to 27.9 points, *p* = 0.003) as well as from POD 6 to POD 9 (27.9 to 30.1 points, *p* < 0.001). The average score of arousal (nervous vs. calm) increased significantly from the preoperative day to POD3 (24.1 to 28.8 points, *p* < 0.001).

### 3.2. Postoperative Complications

Almost half of the patients (*n* = 46, 58%) developed postoperative complications according to Clavien–Dindo classification [32,33] (Table 3). Most of the patients developed complications grade II (*n* = 24, 30% of all complications), followed by grade I (*n* = 12, 14%), grade IIIa (*n* = 5, 6%), grade IIIb (*n* = 2, 3%) and grade IV (*n* = 2, 3%). None of the patients developed grade IVb complications, and none of the patients died postoperatively.

### 3.3. Factors Affecting Postoperative Complications

Factors affecting postoperative complications were analyzed using multivariable logistic regression. Except for a previous cancer diagnosis, no significant influencing factors were found (Table 4). We waived analyses of some pre-existing illnesses (chronic inflammatory bowel disease, diabetes, renal insufficiency), drug consumption, and diet as well as of MDMQ on the sixth and ninth postoperative days, due to the low sample size of subgroups.

MDMQ results of patients with development of postoperative complications did not differ significantly from the MDMQ results of patients without postoperative complications. We found significantly lower wakefulness scores on the sixth POD (26.6 vs. 30.0 points, *p* = 0.038) and a tendency for a lower arousal score arousal on the third POD (27.7 vs. 30.3, *p* = 0.051) in patients with postoperative complications. The results of MDMQ separated by complications are shown in Table 5.

Subgroup analysis of patients with severe complications (Clavien–Dindo grade III, IV, and V; *n* = 9) revealed a significantly lower arousal score on the third POD (25.0 vs. 29.3 points, *p* = 0.030) compared to patients with mild to moderate or without complications. We also found a tendency for a lower mood score on the third POD in patients with severe complications, but the comparison did not reach the level of statistical significance (26.6 vs. 30.0 points, *p* = 0.071).

### 3.4. Influence of Results of Questionnaire on Length of Hospital Stay

Length of hospital stay lasted on average 10 days (range 5 to 22 days). There was no correlation between the length of hospital stay and the MDMQ results. Patients with the development of postoperative complications had a significantly longer length of hospital stay than patients without complications (complication: 10.6 ± 3.4 vs. no complication: 8.6 ± 2.7 days, *p* < 0.001).

## 4. Discussion

Psychological distress is known to affect an individual’s life and is induced by potential life-threatening events such as surgery, but the results of our study suggest that elective colorectal surgery is associated with only moderate emotional strain. The initial hypothesis that preoperative psychological distress measured by MDMQ has an impact on the development of postoperative complications of patients undergoing elective colorectal surgery could not be confirmed by our results. Nevertheless, the results of this trial are interesting, showing early postoperative conspicuous scores of arousal and wakefulness in patients with later development of postoperative complications.

Of all patients eligible, almost 80% of patients were able to complete pre- and third-day postoperative questionnaires, implying a good response to MDMQ. The sample size for evaluation of the primary aim of this trial, the comparison between preoperative MDMQ results and the results on the third preoperative day, was accomplished, but the response at day 6 and day 9 was clearly decreased, making especially the ninth POD hard to evaluate. Therefore, the results might not reflect the reality of the psychological distress of patients at day 6 and day 9. It can also be assumed that patients in poor general or physical condition are not interested in participating in the study. The MDMQ results of our study might, therefore, be better than reality. The main weakness of the questionnaire is the five-point scale implying error of central tendency. However, in a publication by Hinz et al., the MDMQ was told as a good alternative for cross-sectional studies and trend evaluations [30], and it has become a commonly used questionnaire in German-speaking countries [28,29,35]. To our knowledge, this is the first survey evaluating psychological distress measured by MDMQ in surgical patients, which is why empirical data is lacking and a comparison to the previous results of surgical patients is not possible. Comparing the MDMQ results of surgical patients to the results of the normal population, the surgical patients’ mood and arousal before surgery appear to be lower (mood: 29 vs. 30 points, arousal: 24 vs. 28 points) whereas wakefulness is higher (28 vs. 26 points) [27,30]. The results of other trials using MDMQ to capture the mental state of young healthy adults tend to show even higher scores of mood (33–35 points) and arousal (27–30 points) [28,29]. In comparison with the MDMQ scores of the normal population and the results of other trials, our results emphasize a slight preoperative emotional distress in surgical patients. Postoperatively, the mood and arousal scores of surgical patients exceeded the results of the normal population, whereas wakefulness scores tended to be lower than the normal population, especially during the early postoperative course. The postoperative results appear to reflect the feeling of relief after surgery as well as expectable postoperative fatigue. Nevertheless, the transferability of results is limited, as the perception of psychological distress depends on various factors such as education, personality, faith, experience, and origin [36,37,38]. The partially high standard deviation of our results show the broad dispersion of psychological distress in our patients. The individual differences in reaction to stress emphasize the necessity of respect, as it is an individual’s burden.

Generally, the human body is exposed to a massive stress event in case of surgery, and the amount of surgery-provoked distress is known to affect the long-term outcome of patients [1,3,23,39]. To date, little is known about the impact of psychological distress on early postoperative outcomes. In daily clinical routine, many surgeons are aware of the bad feeling before surgery, which is provoked by the assumption that the patients could not be able to recover after surgery due to his/her mental state. Recent research suggests that depression and anxiety are associated with worsening long-term outcomes after surgery. Perski et al. found a significantly higher rate of cardiac events in the 3-year follow-up after coronary bypass grafting in preoperatively distressed patients [39]. McHugh et al. reported that higher preoperative levels of depression and anxiety were associated with lower levels of physical health 6 and 12 months after total hip replacement indicating a poor recovery in these patients [40]. Similar results were found one year after a total knee replacement as the quality of life and function was worse in patients with preoperative higher levels of psychological distress [41]. Amaral et al. divided patients undergoing elective lumbar spine surgery into two groups, one group with mild and the other one with moderate psychosocial problems. They reported an impact of psychosocial distress as patients with higher distress had worse postoperative outcomes [42]. In cancer patients, depression appears to be one of the crucial factors in deciding treatment adherence and cancer-related outcome [43]. It is assumable that the postoperative treatment adherence after orthopedic surgery is decreased in patients with depression, which might be causative for the weak long-term outcome results of patients with preoperative distress. In our trial, almost 8% of the patients showed mood scores lowered by more than two standard deviations. It is assumable that these deviations are not only explained by surgery-driven psychological distress as about 10% of the adult population in Germany suffers from affective disorders [44]. The frequency of distinct psychological distress in our trial is in line with results of other trials showing levels of depression and anxiety in about 10% of patients undergoing urological surgery [9].

Psychological and emotional aspects appear to be neglected in surgical research as only a few publications engage the role of mental health in surgical recovery. The first steps to recognize the impact of surgery-driven psychological distress on the long-term outcome of patients are completed, but further research has to elucidate the biopsychosocial network to potentially enable new approaches to prevent postoperative complications and worse long-term outcomes.

## 5. Conclusions

Pre- and third-day postoperative psychological distress measured by MDMQ did not affect the postoperative complication rate of patients undergoing elective colorectal surgery. However, the results of this trial are interesting, showing early postoperative conspicuous scores of arousal and wakefulness in patients with later development of postoperative complications, which should be addressed in future trials. Additionally, further research has to clarify the impact of psychological distress on long-term outcomes after colorectal surgery.

## Figures and Tables

**Figure 1 jcm-11-02712-f001:**
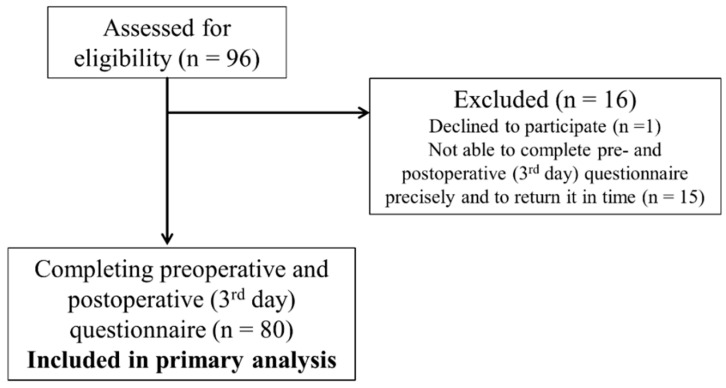
Flow of participating patients.

**Figure 2 jcm-11-02712-f002:**
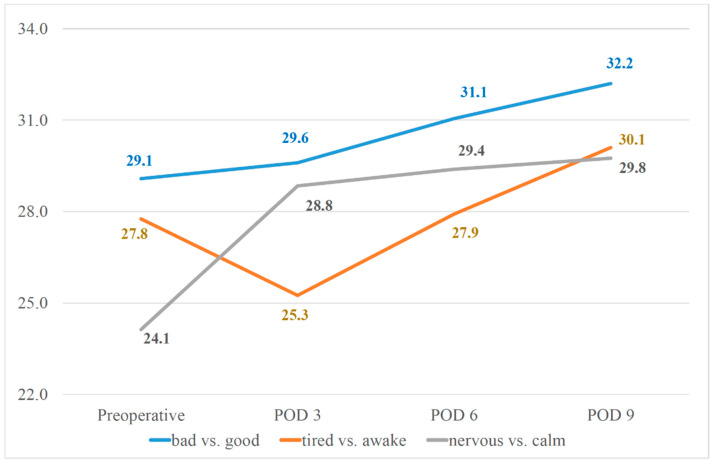
Course of multi-dimensional mood questionnaire results (blue line: mood (bad. vs. good), orange line: wakefulness (tired vs. awake), grey line: arousal (nervous vs. calm); POD = postoperative day). The mood increased continually from the preoperative day to POD9. Patients were the most tired on the 3rd postoperative day and the most nervous preoperatively and calmed down postoperatively.

**Table 1 jcm-11-02712-t001:** Descriptive data of patients completing questionnaire preoperatively and on the 3rd postoperative day (*n* = 80).

	**Mean ± SD**
Age (years)	58.5 ± 16.1
BMI (kg/m^2^)	25.6 ± 5.1
	***n* (%)**
Gender (male/female)	41 (51%)/39 (49%)
Type of surgery	
- Left hemicolectomy	4 (5%)
- left hemicolectomy	30 (38%)
- Removal of the sigmoid colon	28 (35%)
- Gastrointestinal continuity restoration	11 (14%)
- Ileocaecal resection	3 (4%)
- Recreation of illeotransversostomy	2 (2%)
- Subtotal colectomy	1 (1%)
- Segmental bowel resection	1 (1%)
Surgery for the first time	35 (44%)
Surgical access	
- Laparoscopic	49 (61%)
- Laparotomy	31 (39%)
Pre-existing illness	
- Chronic inflammatory bowel disease	9 (11%)
- Current cancer	35 (44%)
- Previous cancer	16 (20%)
- Cardiovascular disease	32 (40%)
- Diabetes	4 (5%)
- Renal insufficiency	4 (5%)
Smoker	
- Yes	14 (18%)
- No	42 (53%)
- Quitted	24 (29%)
Alcohol	
- Regularly	7 (9%)
- Occasionally	48 (60%)
- No	25 (31%)
Drug consumption *	1 (1%)
Diet	
- Omnivore	77 (97%)
- Vegetarian	2 (2%)
- Other	1 (1%)

SD = Standard deviation; * One patient stated regular consumption of cannabis.

**Table 2 jcm-11-02712-t002:** Results of multi-dimensional questionnaire (MDMQ) preoperatively as well as on the 3rd, 6th, and 9th postoperative days.

Day	*n* *	Mood *(Bad* vs. *Good)* Points ± SD	Wakefulness *(Awake* vs. *Tired)* Points ± SD	Arousal *(Nervous* vs. *Calm)* Points ± SD	Overall Score *(All)* Points ± SD
Preoperative	80	29.1 ± 7.1 (range 4–39)	27.8 ± 7.1 (range 8–40)	24.1 ± 6.9 (range 8– 38)	81.7 ± 14.4 (range 53–114)
Postoperative 3rd	80	29.6 ± 6.0 (range 19–40)	25.3 ± 7.5 (range 10–39)	28.8 ± 6.2 (range 12–40)	82.5 ± 14.8 (range 54–104)
Postoperative 6th	62	31.1 ± 5.7 (range 18–40)	27.9 ± 6.4 (range 14–39)	29.4 ± 5.4 (range 17–40)	82.8 ± 13.2 (range 56–108)
Postoperative 9th	20	32.2 ± 4.9 (range 21–38)	30.1 ± 6.3 (range 16–40)	29.8 ± 5.6 (range 15–38)	91.3 ± 13.0 (range 65–110)

SD = Standard deviation; * The missing patients on the 6th and 9th postoperative days were already discharged.

**Table 3 jcm-11-02712-t003:** Frequency and types of complication according to Clavien–Dindo classification [32,33].

Grade	*n*	Type of Complication
I	9	Paralytic ileus (short-time, without nasogastral tube)
	1	Wound infection
	2	Hypokalemia
II *	14	Paralytic ileus (long-time, with nasogastral tube)
	3	Rectal drain
	2	Blood transfusion
	1	Albumin infusion
	8	Need for antibiotic treatment due to…
		- Urinary tract infection (*n* = 6)
		- Increasing inflammatory markers without focus (*n* = 2)
IIIa	2	Hematochezia (with necessity of endoscopy)
	2	Ascites drain
	1	Pleural drain
IIIb	1	Heavy hematochezia (with necessity of endoscopic clipping)
	1	“Acute abdomen” °
IVa	2	Acute renale failure
IVb	0	
V	0	

* Three patients suffered from urinary tract infection and paralytic ileus with necessity of a nastrogastral tube. ° Patient was re-operated due to clinical worsening and development of acute abdomen, no pathology was found intraoperatively and patient recovered without further deceleration.

**Table 4 jcm-11-02712-t004:** Results of multivariable logistic regression: Influence of different factors on development of postoperative complication.

Parameter	Odds Ratio	95% Confidence Interval		*p*
		Lower	Upper	
Age	0.974	0.917	1.033	0.381
Gender	0.385	0.056	2.638	0.337
BMI	1.047	0.870	1.244	0.665
Type of surgery				
- Left hemicolectomy	*	0		1.000
- left hemicolectomy	*	0		1.000
- Removal of the sigmoid colon	*	0		1.000
- Gastrointestinal continuity restoration	*	0		1.000
- Ileocoecal resection	*	0		0.999
- Recreation of illeotransversostomy	*	0		0.999
- Subtotal colectomy	*	0		0.999
- Segmental bowel resection	Reference	Reference		Reference
Surgery for the first time	6.399	0.825	49.614	0.076
Surgical access				
- Laparoscopic	6.523	0.846	50.277	0.072
- Laparotomy	Reference	Reference	Reference	Reference
Pre-existing illness				
- Current cancer	0.228	0.025	2.040	0.186
- Previous cancer	8.077	0.984	66.287	0.052
- Cardiovascular disease	0.778	0.173	3.508	0.744
Smoker				
- Yes	1.331	0.132	13.462	0.809
- No	0.905	0.177	4.618	0.904
- Quitted	Reference	Reference	Reference	Reference
Alcohol				
- Regularly	0.220	0.009	5.279	0.351
- Occasionally	0.795	0.141	4.490	0.795
- No	Reference	Reference	Reference	Reference
MDMQ preoperative				
- Mood	0.942	0.821	1.088	0.434
- Wakefulness	0.945	0.868	1.212	0.430
- Arousal	1.026	0.0784	1.206	0.762
MDMQ 3rd day				
- Mood	0.972	0.784	1.206	0.796
- Wakefulness	1.064	0.925	1.224	0.386
- Arousal	1.156	0.904	1.477	0.248

MDMQ = multi-dimensional mood questionnaire; ***** out of range.

**Table 5 jcm-11-02712-t005:** Results of MDMQ in patients with development of postoperative complications compared to patients without complications.

Dimension of MDMQ	Patients with Complications (*n* = 46) *	Patients without Complications (*n* = 34) *	*p*
Mood			
Preoperative (*n* = 80)	30.2 ±5.5	27.7 ± 8.7	0.373
3rd POD (*n* = 80)	29.0 ± 5.6	30.3 ± 6.4	0.280
6th POD (*n* = 60)	30.6 ± 5.4	31.0 ± 6.2	0.238
9th POD (*n* = 20)	32.0 ± 5.4	33.4 ± 2.5	0.892
Wakefulness			
Preoperative (*n* = 80)	29.0 ± 6.4	26.2 ± 7.7	0.160
3rd POD (*n* = 80)	24.3 ± 7.3	26.4 ± 7.6	0.230
6th POD (*n* = 60)	26.6 ± 6.5	30.0 ± 5.8	0.038
9th POD (*n* = 20)	29.8 ± 6.8	31.5 ± 3.7	0.820
Arousal			
Preoperative (*n* = 80)	24.1 ± 6.7	24.2 ± 7.3	0.904
3rd POD (*n* = 80)	27.7 ± 6.1	30.3 ± 6.1	0.051
6th POD (*n* = 60)	29.3 ± 5.1	29.5 ± 6.0	0.971
9th POD (*n* = 20)	29.3 ± 5.8	31.5 ± 4.7	0.617

* Lower samples sizes on the 6th and the 9th POD due to discharges; MDMQ = multi-dimensional mood questionnaire; POD = postoperative day.

## Data Availability

Raw data are available by the corresponding author on request.

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
