# Peer review of "Surgery as an Emotional Strain: An Observational Study in Patients Undergoing Elective Colorectal Surgery"

_jcm, 2022, doi:10.3390/jcm11102712_

Round 1
Reviewer 1 Report
The authors present a study with an interesting objective, but the study has many shortcomings.
Introduction:
It needs to be expanded, increasing the background knowledge and contextualisation of the study.
Methodology.
The main problem I see is the method used in the study. They use a scale that seems not to be validated and, according to the authors in the discussion, they have not found previous studies where it has been used.
Is the scale used in the assessment of patients validated? This should be reflected in the methodology section. Why was it decided to use this scale?
Who carried out the assessments?
All these data are not reflected or clarified.
Results
It is observed that the number of responses decreases as the days go by, so the results obtained may not reflect the reality of the problems of these patients or the objective to be pursued, given that the sample population is also small.
DISCUSSION
The results obtained are not discussed, nor are the limitations of the study reflected or addressed in detail.
Author Response
Thank you for your time and your thorough review of our manuscript! Your concerns made our manuscript more precise and accurate, which really improved our manuscript. All changes in the manuscript are highlighted in yellow.
The authors present a study with an interesting objective, but the study has many shortcomings.
Introduction:
It needs to be expanded, increasing the background knowledge and contextualisation of the study.
Response: Thank you for your suggestion! We revised the introduction of our manuscript (page 2, line 47 and page 2, line 57).
Methodology.
The main problem I see is the method used in the study. They use a scale that seems not to be validated and, according to the authors in the discussion, they have not found previous studies where it has been used.
Is the scale used in the assessment of patients validated? This should be reflected in the methodology section. Why was it decided to use this scale?
Who carried out the assessments?
All these data are not reflected or clarified.
Response: We are really sorry for this misleading wording! The scale was not used before in surgical patients, but it is commonly used in other, especially German-speaking, trials including healthy participants or the normal population (for example DOI: 10.1038/s41598-020-69280-9, DOI: 10.1186/s40798-021-00325-7 and DOI: 10.1055/s-0031-1297960). We have added “The MDMQ is a commonly used questionnaire in Germany to capture the mental state of a respondent” to clarify this concern (page 2, line 92). The comparison of our results with the results of the normal population can be found in Discussion (page 10, line 235). We also added a few more words about the usage of MDMQ to the Discussion of our manuscript (page 10, line 230).
We moved “The decision to measure mood by MDMQ was driven by the necessity of a standardized, but simple and quick to answer questionnaire to ensure that postoperative patients are not additionally stressed by the questionnaire and are able to finish it in a few minutes” from Discussion to Methods to explain the rationale to use the questionnaire (page 3, line 102).
The results were evaluated by three authors with the help of an evaluation template (page 3, line 106), which is made by Testzentrale, Göttingen, Germany.
Results
It is observed that the number of responses decreases as the days go by, so the results obtained may not reflect the reality of the problems of these patients or the objective to be pursued, given that the sample population is also small.
Response: Yes, you are right that the decreasing number of responses limits the results, but this was a secondary target of our trial. We added this aspect also to the Discussion (page 10, line 224). The primary aim of the study was the association of the preoperative results and the development of postoperative complications according to Clavien-Dindo classification. The planned sample-size is reached for the primary aim.
DISCUSSION
The results obtained are not discussed, nor are the limitations of the study reflected or addressed in detail.
Response: We revised “Discussion” and expanded the limitations as well as the comparison with other results.
Reviewer 2 Report
The article is interesting and it discussed about whether mood would affect post-op complications. However, it would be more informative if the authors can talk about the mood changes in patients with complications. Moreover, the authors can add information about any patients being admitted to ICU care. Patients needed ICU care would probably have lower mood.
Author Response
Thank you for your review! Your concerns were very valuable as they uncovered a really displeasing data base failure regarding the amount and type of postoperative complications.
The article is interesting and it discussed about whether mood would affect post-op complications. However, it would be more informative if the authors can talk about the mood changes in patients with complications. Moreover, the authors can add information about any patients being admitted to ICU care. Patients needed ICU care would probably have lower mood.
Response: We revised all of the complications results thoroughly and considered your concern to go more deeply into the mood of this patients (see Table 5 and page 9, line 201). Unfortunately, we were not able to add information about the relation between mood and the ICU stay of our patients as the time on ICU was not captured in our trial. This is an interesting aspect, which should be definitely addressed in further trials.
Round 2
Reviewer 1 Report
Although the authors have made an effort to improve the manuscript, the introduction remains sparse, the data they have provided have not improved the manuscript in this respect. The introduction lacks data to contextualise the study. On the other hand, the main problem I see is the methodology of the study, the scale used is not clear whether or not it has been validated, which could lead to biases in the results obtained.
Furthermore, although it is discussed in the discussion section, 'It is observed that responses decrease as the days go by, so that the results obtained may not reflect the reality of the problems of these patients or the objective to be pursued, given that the sample population is also small
Author Response
Thank you again for reviewing our manuscript!
We revised the introduction again focusing more on the relation between body and mind to emphasize the role of mental health for recovery in surgery.
I am sorry as my first response might be misleading to you. The questionnaire is a well-validated questionnaire, which is used for more than 20 years. We have added another reference to constrain the validity of the survey. Before study onset, we had a discussion with the head of the psychosomatic department of our university medical center, who is an international recognized expert, to choose a suitable questionnaire for our aim. We followed his recommendations.
Thank you again for pointing out the aspect that the results might not reflect the psychological distress of patients at day 6 and day 9. We added the limitation to "Discussion" (page 9, line 227).
Reviewer 2 Report
The author has made lots of changes and the article is well written. Although it is a negative study (as expected), I do think it's an interesting article and worth for publication.
Author Response
Thank you again for your evaluation! We are happy about your decision.
Round 3
Reviewer 1 Report
The authors have corrected the details discussed in the introduction, but the main problem persists, and cannot be remedied because the sample used is too small. Moreover, it is observed that the responses decrease as the days go by, so that the results obtained may not reflect the reality of the problems of these patients or the objective pursued, given that the sample population is also small.